# Towards Intraoperative Tissue Characterisation with Industrial Precision LiDAR

Noan Le Renard
*Electrical and Electronic Engineering*
*Imperial College London*
SW7 2BT, UK
nml18@imperial.ac.uk

Steven S. Wong
*Electrical and Electronic Engineering*
*Imperial College London*
SW7 2BT, UK
steven.wong15@imperial.ac.uk

Stamatia Giannarou
*Surgery & Cancer*
*Imperial College London*
SW7 2BT, UK
stamatia.giannarou@imperial.ac.uk

Daniele Faccio
*Physics and Astronomy*
*University of Glasgow*
G12 8SU, UK
Daniele.Faccio@glasgow.ac.uk

Timothy G. Constandinou
*Electrical and Electronic Engineering*
*Imperial College London*
SW7 2BT, UK
t.constandinou@imperial.ac.uk

Jinendra Ekanayake
*Electrical and Electronic Engineering*
*Imperial College London*
SW7 2BT, UK
j.ekanayake@imperial.ac.uk

*Abstract*—There is a recognised need for real-time, point-of-procedure tissue identification during resective tumour surgery; this is made more significant by the need to account for the tissue shifting during tumour surgery in areas such as the liver and brain. This challenge with tissue mobility, deformation and 'shift' leads to the preoperative imaging which is currently used to localise tumours, such as MRI or CT scans, being rendered inaccurate or misleading.

In this work, we explored the use of an industrial precision handheld Laser Line Probe (LLP) with 25-micron accuracy to extract tissue viscoelastic information, with the goal of identifying healthy and cancerous tissue in real-time. This is anticipated to contribute to significantly improved surgical outcomes, with scalability to resource limited and technology sparse environments.

Simulation of intraoperative palpation was robustly paired with the LLP scanning and during direct probing of high-fidelity tissue models. We obtained point cloud scans which emulated time-series data, with the scan line characterising tissue deformation in 3D. By extracting physical and 3D point cloud features, we trained a Random Forest model capable of classifying and differentiating biophantom and nonorganic samples with a 96% 10-fold cross-validation accuracy.

*Index Terms*—Tissue deformation, Machine Learning, Medical Imaging, Point Cloud Data, Laser Line Probes, Viscoelasticity.

## I. INTRODUCTION

Pre-operative imaging, including MRI and CT, are the current standard of care for visualising, localising and delineating intrinsic tumours such as liver or hepatocellular carcinoma, and brain tumours such as gliomas. This step is fundamental in surgical planning as it provides non-invasive three dimensional visualisation for guiding targeted surgery by localising the tumour and helping to establish its size, depth, invasiveness and boundaries. However performing the surgical opening of the abdomen or the skull produces movement of organ of interest, which becomes progressively worse as tumour is manipulated and removed. With the liver this arises as a result of changes in intrabdominal fluid and pressure. Within the skull, this is due to fluid shifts in the cerebrospinal fluid (CSF) filled intracranial spaces as well as the folded nature of the brain, which leads to the brain deforming unpredictably. This phenomenon of tissue deformation is made more significant as surgical resection of the tumour is undertaken due to swelling, gravity, tumour resection, fluid drainage, and other factors. It is particularly relevant when attempting to determine tumour borders next to functional regions during the tumour resection. It is crucial to establish accurate tumour boundaries, as this enables more confident and complete tumour removal while avoid digressing into healthy adjacent tissue. Image guided tumour removal is directly linked to better surgical outcomes [1]. The goal is to achieve a balance between resection of tumour and preservation of functional tissue as overly aggressive resection can lead to functional deficits, while insufficient removal increases the risk of tumour recurrence.

Surgeons predominantly rely on pre-operative imaging to approximate tumour margins; this becomes compromised due to highly variable tissue deformation ranging from a few millimetres (mm) to over 25 mm and has been demonstrated to be patient-specific and non-linear [2]. Existing intraoperative imaging solutions such as ultrasound (USS) are highly user-dependent and challenging to implement due to limited soft tissue contrast, artefacts, inferior image quality as compared to MRI and CT, and a lack of standardisation due to variability in equipment, techniques, and interpretation. Other solutions include labelling with compounds such as 5-aminolevulinic acid (5-ALA). This drug is administered prior to patients undergoing neurosurgery to enable fluorescence guided tumour resection using specially equipped microscopes [3]. However, this method is subject to risks and a side effect profile . Further, tumour interfaces are not well delineated using luminescence, with clinicians being required to repeatedly change views with specialised grounded microscopes. These are expensive and can be a challenge to acquire in resource-constrained environments. Finally, surgeons use direct palpation to distinguish pathological from healthy tissues during intraoperative proce-

dures [4]. Although palpitation is both a fundamental surgical soft skill and has demonstrated effectiveness, it remains a subjective experience lacking quantitative interpretation or the opportunity for clinical documentation.

Tumours have distinct physical properties of tumours which set them apart from healthy tissue. They are physically identifiable by their compressibility and stiffness as compared to non-pathological tissue, i.e., viscoelastic properties [5]. As such leveraging these physical disparities presents an opportunity to distinguish tumours from healthy tissue, enabling a potential advance in real-time intraoperative tissue differentiation linked to the surgeon's intuitive haptic interaction with the tissue.

The aim of this study was to explore the feasibility of employing a handheld light detection and ranging (LiDAR) device for the extraction of tissue information. This technology allows for the real-time capture of sub-millimetre accuracy for tissue scanning. This is directly relevant to the proposed application at the point of care. It was hypothesised that changes in point cloud data during direct tissue palpation allow identification of the tissue based on its viscoelastic properties. The overarching goal is to identify healthy and cancerous tissue margins in real-time at a level of accuracy which is currently not achievable with existing approaches. Further, the handheld implementation enables a simplified surgical workflow which can be scaled to resource limited locations, avoiding the need for complex, expensive and large technologies e.g. one or all of the following: intraoperative MRI, neuronavigation MRI stacks and screens, intraoperative ultrasound stacks and blue-light microscopes for 5-ALA. There is precedent for the use of handheld devices during tumour removal, such as direct cortical stimulation, neuronavigation MRI-linked handheld pointers, optical coherence tomography [6] or gamma counting probes combined with radiolabelling of tumour tissue [7] [8]. However, the application of a high precision Laser Line Probe (LLP) in this context provides a high-resolution, real-time visualisation approach which is label free. It is unique based on the surgical and technological review by the authors which include experts from engineering, biophotonic medical devices and neurosurgery.

## II. DATA COLLECTION

### A. Technology

LiDAR technologies function by emitting a targeted laser and measuring the time it takes for the reflected light to return to a receiver, thereby calculating the distance to an object or surface. A LLP employs this concept but emits a narrow line of laser, known as a scanline, instead of a single point. This allows for more precise and simultaneously more widely encompassing measurements, particularly suitable for smaller-scale projects compared to traditional LiDAR applications. As a result, LLPs are commonly used in fields such as manufacturing, forensics, and archaeology. Industrial-precision LLPs, i.e. micron level accuracy, have not been previously used in surgical imaging contexts, setting the stage for this feasibility study.

Exposure to lasers, such as that utilised in LiDAR technology potentially posed risks to human tissue due to heat generation, photochemical reactions, and ionization. However, the FARO LLPs use a 450/635 nm Class 2 laser, which does not damage the skin or the eyes (unless deliberately directed at the eye for an extended duration), with an output power ¡1mW, and with a Maximum Permissible Exposure of 2.55 mW/cm$^2$ [9]. As such, it is assumed that this technology will not damage human tissue and is safe.

### B. Hardware

The project utilised the FARO® Quantum Max ScanArms with Multiple Laser Line Probes [10] (see figure 1). This 6-axis robot, which was ground-mounted using a tripod, was equipped with the xS laser line probe capable of acquisition rates of up to 1,200,000 points per second, an accuracy of $25\mu$m and a minimum point spacing of $30\mu$m. The LLP successfully achieved scanning at working distance of 10-25 cm from the target surface. The maximum point per scanline was 4000 with a maximum scan rate of 600 Hz. The provided FARO® RevEng$^{TM}$ software was used with a physical connection to capture data, calibrate/configure the robot, and export models from the ScanArms.

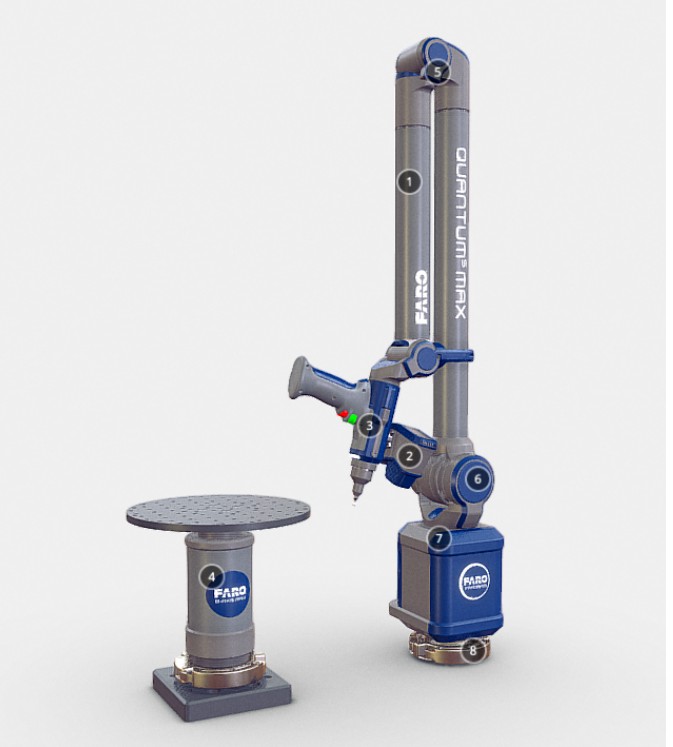

Fig. 1. FARO® Quantum Max ScanArms with Multiple Laser Line Probes [10].

For initial use, the LLP necessitates a calibration process for its sensors. This is done by performing predefined scans at fixed angles in sequence according to the software's instructions. This calibration sequence was time-consuming and challenging during the initial attempt, as the tolerances for

sweep distance and angles were very tight. Once calibrated, the robot does not require re-calibration unless there is hardware malfunction or a change in probe type.

### C. Inorganic phantoms and Biophantoms

Due to the lack of access to *in-vivo* samples, we used a range of standard inorganic phantoms to benchmark the LLP device including a silica gel bag, synthetic tarp and a dessicant bag. We additionally used biophantoms. Biophantoms are intended to be high-fidelity artificial models designed to simulate the physical and optical properties of biological tissues. There were used to benchmark the LLP's performance and to collect data on our methods. The ones used were as follows:

- Liver phantom (see figure 2): Custom-made phantoms by the Hamlyn Centre, Imperial College London. Contains structures within that simulate tumorous areas.
- Abdominal organ phantoms (Small and large intestine, gall bladder): Custom-made phantoms by Prof. Fernando Bello, Chelsea and Westminster Hospital.
- Brain phantom: Manufactured by Organa Technology Ltd from Case 7 of the RESECT dataset, chosen due to the large size and superficial location of the tumoural mass.

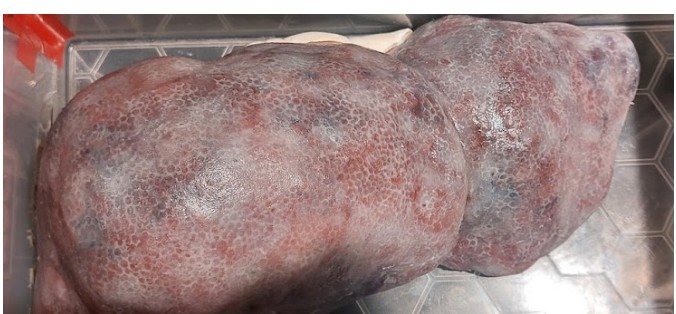

Fig. 2. Pathological liver biophantom used for data collection.

Original MRIs were used to manufacture the tissue models. To enable different physical properties of a 'tumour' component within the model, it was necessary to perform a separate segmentation of the tumour from the rest of the tissue. The segmentation step has been performed using Slicer3D by labelling reference points belonging to the tumour and then using seed-growing segmentation. This output was then exported from NIFTI to STL format for the 3D printing of the manufacturing mould. The phantoms were made out of a unique material that mimics mechanical properties relevant to surgery (stress-relaxation and fracture toughness) using a proprietary manufacturing process. This enabled the model to simulate how the tissue would deform during a surgical procedure. Finally, the tumour mass contained in the phantoms were manufactured using a different stiffness and toughness compared to the rest of the tissue. The tumour was also differentiated by colour to help locate it visually.

The brain phantom was kept in ionised water for preservation purposes.

### D. Methodology

To identify tissues, with the overarching goal of extending this to real-time tumour-tissue interface identification, intraoperative direct palpation, as implemented with a surgical probe, was quantitatively measured using the LLP.

Using a probe to palpitate a with the LLP held at a constant height yielded a point cloud of a 3D depression of the probe into the tissue. The scanlines directly reflected the evolution of the pressing motion over time, generating the pseudo-time series data. The visualisation and manipulation of these scans was done using the Open3D library [11].

Initial testing on various objects showed that, even with the same amount of palpitation pressure, the physical characteristics of the depression and point-cloud model was varied. This is demonstrated in the following figures 3,4.

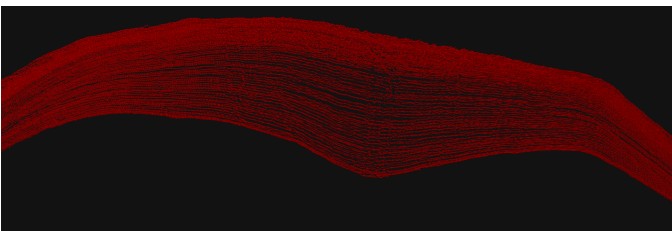

Fig. 3. Pre-processed point cloud model of the depression caused by controlled palpation on a pathological liver biophantom.

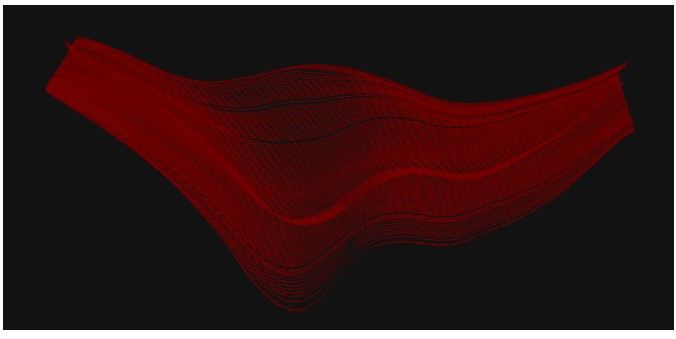

Fig. 4. Pre-processed point cloud model of the depression caused by controlled palpation on a synthetic tarp.

### E. Dataset

A comprehensive dataset of palpation data was collected from a wide corpus of subjects, comprising approximately 25 samples each as follows:

***Inorganic Phantoms:***

- Dessicant bag
- Synthetic Tarp
- Silica Gel Bag

***Biophantoms:***

- Liver (Apex of Left/Right Lobe, Falciform Ligament)
- Liver 2 (from a different 'abdomen')
- Gallbladder
- Small Intestine

- Large Intestine

***Inorganic Objects:***
- Anti-Static (ESD) Mat
- Plastic composite Table

The dataset presented variation in physical proprieties that spanned between tumour and non-pathological tissue, i.e., based on compressibility and stiffness. This included scans from varying locations of the same biophantom as shown in figure 2. A successful tissue identification solution needed to accurately discern between these diverse samples. As anticipated, the models exhibited variations in physical appearance and characteristics across the different subjects.

The absence of predefined patterns and structures in the dataset made the construction of a rule-based approach to this identification problem unfeasible. As such, a machine learning approach was considered the most relevant and principled approach, taking into account practical and operational requirements.

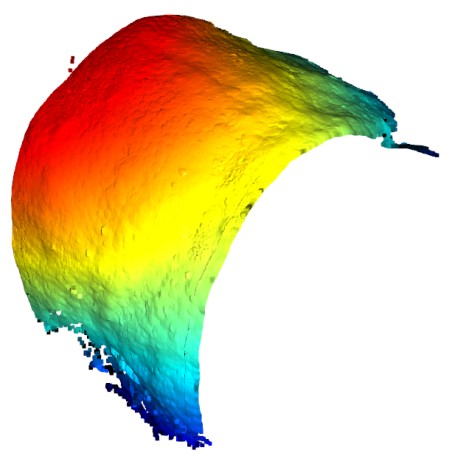

Fig. 5. Point-cloud scan using the LLP of the Apex of the right lobe of the Pathological liver biophantom.

High quality images were obtained from the liver and abdominal biophantoms (see figure 5). However some challenges were encountered during the data capture phase The LLP's efficiency varied depending on the surface properties. Glossy, transparent, or translucent surfaces produced reflection and/or distorted the laser, causing noise or doubling the point interpolation. This was especially relevant for the brain biophantom which was partially suspended in water. The laser scattering caused noise and unwanted artefacts in the point cloud model, rendering this subject unusable (see figure 6).

Other factors included surface colour. Dark surfaces absorbed more light, dampening the laser intensity and interfering with distance measurements. Further the imaging the object's edges required addressing the potential for the laser to 'boomerang'. To address these issues, the following scanner configurations were implemented:

- CLR profile: HDR

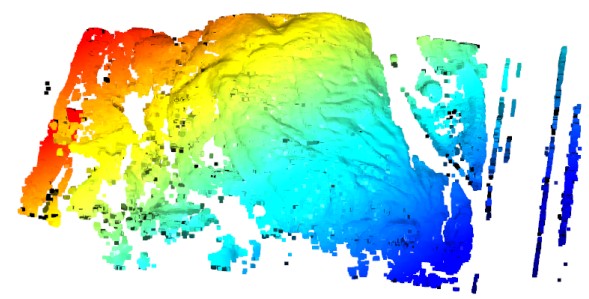

Fig. 6. Point-cloud scan using the LLP of brain biophantom.

- Scan density: 1:1
- Movement filter: 0.1700 (lowest)
- Reflection Filter: 0 (lowest)
- Angular Filter: 80 (highest)
- Capture Method: Raw Data

Typical point cloud post-processing methods such as voxel downsampling, Poisson Surface Reconstruction, uniform sampling, and smoothing were also tested. However, alongside a large computational overhead, these either decreased model performance or varied in a statistically insignificant way. As such, bespoke pre-processing methods were developed.

*F. Verification*

Ensuring data reproducibility was a primary concern due to potential variations in applied force and probe location. To address this, an object-to-object comparison of the different depression depths obtained by the samples was conducted. The variation in the local minima and maxima of the depression were computed using RevEng's measuring tool, and distances from the lowest to highest depression points were recorded for each model. Across objects, distances consistently fell within 0.5 mm, indicating satisfactory reproducibility. While the surgeon's dexterity in using a probe may enhance precision in an intraoperative setting, ideal reproducibility might be best achieved with an automated placement of the mechanical probe for each scanning run.

Probe location variability during data collection was successfully addressed by using LLP's laser guides which ensured the probe was held within the optimal scanning range and maintained in 2 axes for proper data acquisition.

### III. Feature Selection and Extraction

Given the infeasibility of visual/manual inspection within our dataset, the extraction of features was necessary to offer a basis for comparison of the training of machine learning models. Features were extracted based on the model's physical, geometrical, and statistical properties. A bespoke algorithm was developed to accurately measure the depression depth, leveraging the constancy of pressure application across samples to glean essential information regarding tissue elasticity.

**Algorithm 1** Depression Depth Algorithm.

1: Find the mean x-value **mean_X** of the point cloud.
2: For all points in the **mean_X** column, find the lowest y-value, **lowest_y_mean_x**.
3: Define a KDTree and find the point, **local_minima** within a predefined range with the lowest y-value in a nearest-neighbour search around **lowest_y_mean_x**.
4: Generate a new point cloud from the previous one, removing any point lower than **local_minima**. This now allows for the extraction of features only from the depression and the removal of artefacts and outliers from the scanning process.
5: Using the average position resulting from the sampling of **local_minima** $\alpha$ times to reduce noise, find the point **maxima** with the highest y-value in the same column.
6: Using the average position resulting from the sampling of **maxima** $\alpha$ times for the same reason, calculate and return the absolute distance between both sampled **maxima** and **local_minima**.

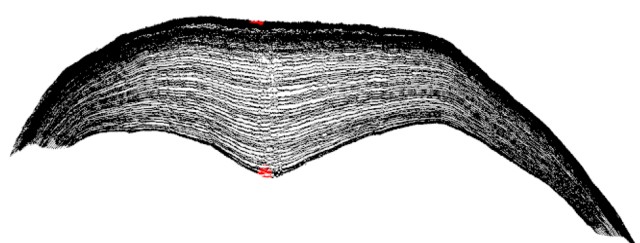

Fig. 7. Post-processed point cloud model of the depression caused by controlled palpation on a pathological liver biophantom, with the local minima calculated from the Depth Algorithm 1 (highlighted in red).

To ensure that a point is found when the associated highest or lowest y-values are searched for a given point, Numpy's `isclose()` function was used with a relative tolerance parameter of 1e-04 and an absolute tolerance parameter of 1e-07.

Harshit et al. [12] present an overview of the extraction of geometric features from 3D point cloud data and review their successful uses in previous research. Using the features detailed with 3 principal Eigenvalues and Eigenvectors (called $\lambda_1, \lambda_2, \lambda_3$ with $\lambda_1 > \lambda_2 > \lambda_3 > 0$) computed using PCA on the point cloud yielded statistically significant results.

After testing features both bespoke and from literature and using the correlation matrix of the features to remove multicollinearity, the final features employed were:

- **Depression Depth** (c.f.1)
- **Omnivariance**: $(\lambda_1 \cdot \lambda_2 \cdot \lambda_3)^{1/3}$, quantifies the overall variance or spread of points in all directions within a region.
- **Anisotropy**: $(\lambda_1 - \lambda_3)/\lambda_1$, describes how much a particular region within a 2D space deviates from being isotropic, i.e. having the same properties in all directions.

- **Planarity**: $(\lambda_2 - \lambda_3)/\lambda_1$, describes how flat or planar a surface or region is within a 3D space.
- **Sphericity**: $\lambda_3/\lambda_1$, assesses how closely a region within a point cloud resembles a sphere or a perfectly round shape.
- **Root Mean Square Roughness**: average profile height deviations from the mean line. Defined as such by ISO 25178, with with $l_r$ the total amount of points:

$$\mathbf{Rms} = \sqrt{\frac{1}{l_r} \int_0^{l_r} z(x)^2 dx} \tag{1}$$

- **Skewness**: the measure of the asymmetry of the profile about the mean line.

$$\mathbf{Rsk} = \frac{1}{\mathbf{Rms}^3} \left[ \frac{1}{l_r} \int_0^{l_r} Z^3(x) dx \right] \tag{2}$$

- **Kurtosis**: the measure of the peakedness of the profile about the mean line.

$$\mathbf{Rku} = \frac{1}{\mathbf{Rms}^4} \left[ \frac{1}{l_r} \int_0^{l_r} Z^4(x) dx \right] \tag{3}$$

- **Convex Hull Volume**: the volume of the bounding box of the smallest convex polygon that encloses all of the point cloud's data.
- **Explained Variance Ratio** (2-tuple): computed by fitting the data to a PCA model with 3 components.
- **Second Moment** (3-tuple): statistical second moment computed by considering the depression to be a statistical distribution.
- **Aspect Ratio**: ratio of the difference between the maximum and minimum x and y values of the post-processed model.

## IV. MACHINE LEARNING

### A. Approach

In view of the observation that the data samples were manually labelled during collection, the classification problem was treated as a supervised multi-class classification task. Given the lack of precedent, an exploratory approach was necessitated, testing a range of commonly used machine learning algorithms to identify the most suitable method for the given use case.

### B. Preprocessing

Data outliers were removed in the scanning step, through a combination of manual and programmatic intervention. On a per-label basis, the features were plotted, revealing a naturally occurring normal distribution attributed to inherent noise. Points presenting extreme deviation from the norm ($> 3$ standard deviations) were manually inspected. Their models all contained scanning artefacts from human error (e.g. the work surface was captured due to a lapse in clip plane configuration), which consequently created outliers after the feature extraction. Such examples were rare, but had to be removed to allow the model to effectively generalise.

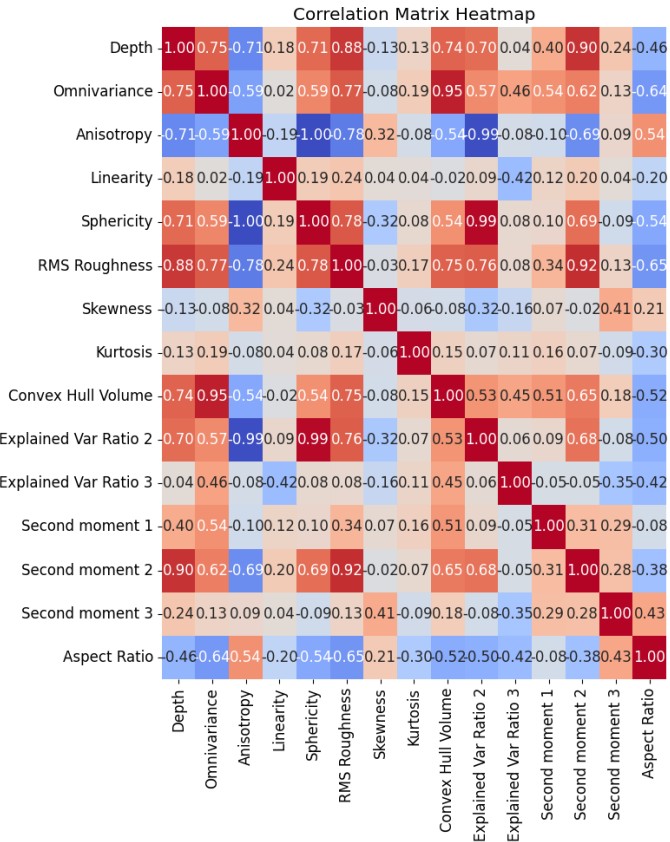

Fig. 8. Correlation matrix of the features used for machine learning.

The features were scaled using `StandardScaler()` and the labels were encoded using `LabelEncoder()`. Both an 80/20 training/test split and 10-fold Cross Validation were used for training.

### C. Models

Different branches of machine learning methods were evaluated, these include: Non-parametric, linear models, ensemble methods, boosting, probabilistic modes, kernel methods, decision trees and neural networks.

The selected models from these branches have demonstrated high performance in scenarios with limited sample sizes, as is the case in use case [13]. These models are as follows:

- Random Forest Model.
- K-Nearest Neighbours Model (with K=5).
- Linear discriminant analysis (LDA).
- XGBoost.
- Decision Tree Model.
- Naive Bayes Classifier.
- Support Vector Machine (SVM) Model.
- Logistic Regression Model.
- Simple Neural Network consisting of 2 sequential layers of 64 and 32 neurons with ReLU activation, and a final layer with Softmax activation, trained using an Adam optimiser and Sparse Cross-entropy loss.

State-of-the art models in this field usually include Deep Neural Networks, such as architectures like Convolutional Neural Networks (CNNs) and Recurrent Neural Networks (RNNs), which are highly effective for complex multi-classification on high-dimensionality data. However, they require large datasets to avoid overfitting and to fully leverage their capabilities. Thus given our limited amount of data, we opted to omit them from our testing.

### D. Implementation

The `scikit-learn` Python library [14] was employed for the training and evaluation of the models. This library offers direct classes for the implementation of various models, such as `KNeighborsClassifier`. The `fit` method was employed to train the models, and the `predict` method was used to generate model output.

The balanced dataset enabled the use of the `accuracy_score` method to evaluate the model's performance. Additionally, the `classification_report` function was employed to generate metrics such as precision, recall, and F1-score. However, the utility of these metrics was limited due to the consistently high accuracy scores obtained.

## V. RESULTS

TABLE I
MACHINE LEARNING RESULTS WITH 10-FOLD CROSS VALIDATION AND 80/20 TRAIN/TEST SPLIT

| Machine Learning Model | Train/Test Split Accuracy (%) | 10-Fold Cross Validation | |
|---|---|---|---|
| | | Accuracy (%) | Std Dev |
| **Random Forest** | **98.4** | **96.4** | **2.3** |
| KNN (K=5) | 96.8 | 89.9 | 4.4 |
| LDA | 98.4 | 92.3 | 5.9 |
| XGBoost | 96.8 | 96.1 | 1.3 |
| Decision Tree | 95.2 | 94.8 | 2.9 |
| Naive Bayes | 96.8 | 95.1 | 2.1 |
| SVM | 98.4 | 90.6 | 4.8 |
| Logistic Regression | 100 | 91.6 | 4.6 |
| Neural Network | 100 | 63.4 | 7.3 |

Logistic regression and simple neural network achieved 100% accuracy (see table I), which required assessment of potential overfitting to the dataset. To address these concerns, K-Fold Cross Validation results were considered. As anticipated, certain models, particularly the neural network, exhibited decreased accuracy, However, two models, Random Forest and XGBoost maintained exceptionally high accuracy. Analysis of the confusion matrix of the highest performing model, Random Forest with 96% 10-fold cross-validation accuracy, revealed only one misclassification, occurring between samples (as seen in the confusion matrix 9). These were sampled from the same biophantom and had similar physical properties, making it a potentially explainable statistical error. This model's average training time was 0.1 seconds, with an average inference time

of 0.005 seconds (AMD Ryzen 5 5600G @ 3.90 GHz, 16GB DDR4 RAM).

This model exhibited comparable performance across other standard machine learning classification metrics:

- AUC-ROC: 0.999
- F1-Score: 0.964
- Precision: 0.965

The liver biophantoms with incorporated tumour, offered a comparable variation in physical properties to real hepatocellular carcinoma, and had each of its 25 samples from 3 areas of different tumour pathologies correctly classified.

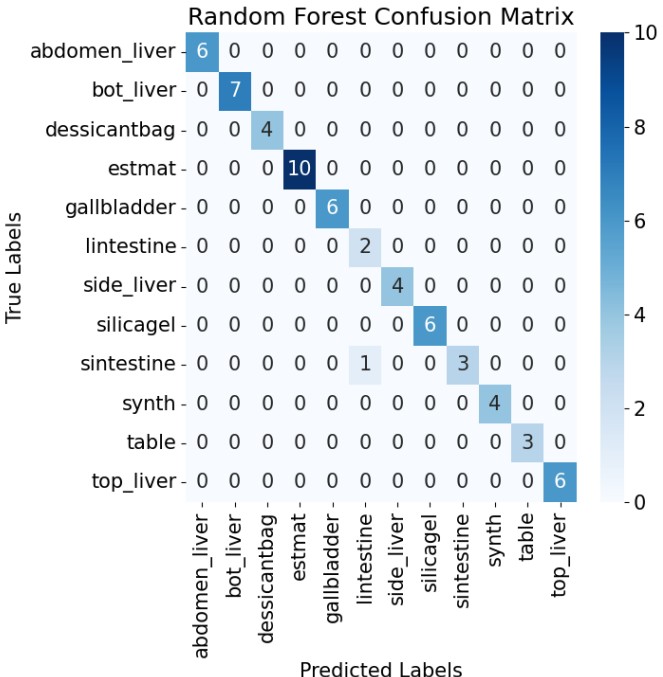

Fig. 9. Confusion matrix of the Random Forest Model from the test/train split
.

This machine learning approach allowed for the creation of a model capable of classifying point cloud models of different samples using the features extracted. The robustness of the model was validated using 10-fold cross-validation, and can now substantiate the significance of the selected features. The computational efficiency of the feature extraction alongside the low inference time of the model allows for real-time data classification with a prediction accuracy that can be continuously improved with each scan.

In light of the model's ability to accurately generalise to unseen data, including the identification of different levels of pathologies within the same biophantom, it is conceivable that with access to a high-quality *in/ex vivo* tissue dataset containing labelled healthy and pathological samples, similar features could be extracted. This would enable the training of a similar machine learning model capable of interoperability distinguishing between healthy and tumour tissues from a handheld device, potentially advancing diagnostic capabilities in a range of point-of-care and point-of-procedure settings.

## VI. FEASIBILITY OF LLP USE

Based on this scanning experience with the FARO LLP, we conclude that it is fully feasible to develop this approach towards a clinical setting. The LLP was manoeuvrable and provided consistent high-quality scanning results. The diverse configurations offered by the device enhance its versatility, rendering it suitable for various specialised clinical settings.

LLP technology presents a potential economic advantage, as for this use case, the technology is holistically much cheaper than traditional medical imaging modalities such as CT and MRI. The relatively low hardware requirements for data collection and model execution make it conceivable that this technology could be open-sourced to improve inference quality.

Logistically, the LLP is suited for both operating theatre and laboratory use. It can be operated from a robotic arm, and easily stowed without special handling requirements. The presence of a surgical lighting system would not outshine the Class 2 laser.

A learning curve exists for acquiring high-quality scans with the machine. Both the calibration process and identification of the optimal scanning settings is a labour-intensive, time consuming process, albeit of low-frequency. Maintaining the optimal distance to the subject whilst performing sweeps corresponding to its geometry requires both meticulous attention and dexterity. This would be best achieved by an automated approach. Regardless, with the set-up completed, the capture of palpation samples for either training machine learning models or intraoperative inference is efficient, only taking a few seconds, and could be automated to generate large datasets.

## VII. CONCLUSION

Tumours present varying physical properties compared to non-pathological tissue. These were captured by an innovative technique utilising a LLP to enable the quantification of intraoperative palpation by measuring tissue depressions induced by probing. This method demonstrated consistency and reliability, resulting in the construction of a comprehensive dataset.

Extracting features from the physical, geometrical, and statistical properties of our data and training a gamut of machine learning models yielded an accurate and robust Random Forest machine learning model achieving 96% 10-fold cross-validation on multi-class classification. This success underscores the model's ability to identify between tissues, including the differentiation between different tumoural pathologies and non-pathological tissue on a liver biophantom.

The utilisation of the FARO® Quantum Max ScanArms with Laser Line Probe in a simulacrum of an intraoperative clinical setting has been demonstrated to be feasible, warranting further investigation. The probe was intuitive, provided consistent high quality scanning results, and could be configured to perform in very specific environments such as the operating theatre. With appropriate configuration, clinician

training, and physical accommodations, integration of this probe into surgical workflows appears feasible.

## VIII. Further Work

Hardware improvements by way of attaching an automated robot haptic probe to the LLP, synchronised to the data collection process could enhance consistency, ease of use, and the model's accuracy. By automating the palpation process, the standardisation of the application of pressure during data collection would reduce variability, thus improving the reliability and confidence intervals of the acquired data.

Retesting the methods described above with timestamped labelled *in/ex vivo* healthy and pathological tissue data would increase clinical application and relevance.

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

## Appendix A
### Standard operating procedure

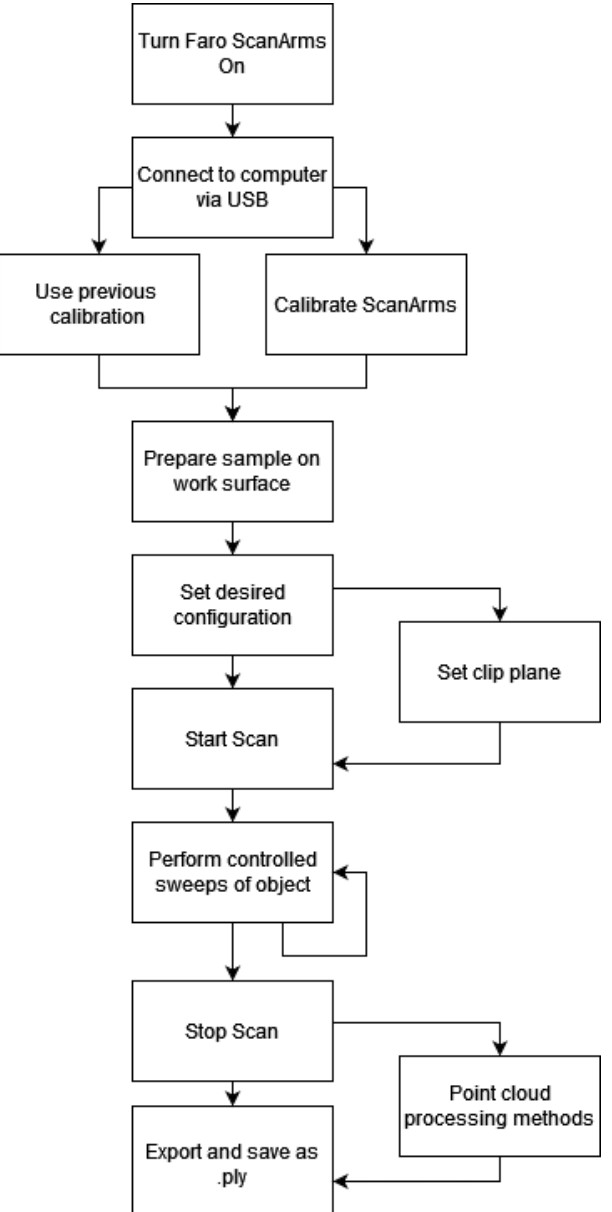

Fig. 10. Standard operating procedure for collecting data using the Faro LLP.