# OpenReview forum: "Towards Intraoperative Tissue Characterisation with Industrial Precision LiDAR"
_IEEE.org/EMBS/BHI/2024/Conference — IEEE BHI'24_

### Official Review · Reviewer_RkN5 · 2024-08-09
**Very promising paper, but results are hard to interpret and support the conclusions presented**

**Overall Rating:** 6
**Confidence:** 2

**Other Quality Metrics:**

● Clarity of writing- Good
● Clinical Significance- Excellent
● Methodological Novelty- Great
● Experiments and Results- Fair

**Questions For The Authors:**

•	Viscoelasticity measurements are mentioned in the introduction. However, from the features extracted, it appears that stiffness is not calculated?
•	Related to viscoelasticity how do you deal with depth-wise variability in stiffness? In ultrasound, you get stiffness as a function of depth, whereas from my understanding this method gets a more global surface stiffness assessment? Would this hinder performance or applicability in the clinical context?
•	It is mentioned in the dataset section that colour/reflection/transparency are issues. If this is the case, how do you know you are not detecting surface property differences instead of stiffness properties? Given that the clinical use-case is presumably on tissues with similar surface properties but different stiffness properties, the current model evaluation setup does not seem appropriate since the multi-class classification has diverse surface properties.
•	Is a precise depression depth or pressure required? I see that depression depth is a feature, but uncertainty if pressure is controlled. If it is controlled, how would a surgeon use this system? Are they applying the pressure? If pressure is not controlled, then is it measured?
•	For the feature selection and extraction section Algorithm 1- what is the x-axis and y-axis referred to here?
•	I may have misunderstood, but it appears that measurements are done from a single depression? If this is the case, how long does this process take? Can it only be automated? How feasible is it in a use-case where multiple regions would want to be measured?
•	A figure may assist with visualisation of the process. As I am unclear how the setup is in relation to pressing the tissue at the same time as shining light and recording change in depth and then in relation to if surgical instruments were present.
•	In Table 1- confused about train/test split accuracy and 10-fold results? As in, how is overfitting the rationale from the train/test split results when the test set is independent of the train set?
•	Figure 9 results, the number of samples appears low from my understanding of the number of samples obtained for each class? Is this cross-validation results? Should it not be around 25 samples?
•	Figure 10, once calibration and setup is done, how long is the process of getting one sample of data?

**Strengths:**

Very interesting topic with a unique technology presented. If the results can be presented more concretely to support the conclusions presented in the paper, then this is a very impactful paper.

**Summary Of The Paper:**

Presents tissue classification based on LLP-based recordings after pressure is applied to phantom tissues.

**Weaknesses:**

Major Comments:
•	Given one of the aims is to identify tumour vs normal tissue results surrounding this would be beneficial. Currently from my understanding of Fig 9 and Table 1 results are for multiclass classification. However, this is already a very diverse of tissues which would not be relevant within a surgery that is generally looking at a subset of tissue types within that area and is more interested in which is tumour and which is normal.
•	Given phantom models are used, it appears to be possible to take a step back from classification. Instead, you could assess the accuracy of stiffness measurements, or looking at the extracted features could confirm if the measured depression depth is actual depression depth, that the anisotropy measure is correlated to the level of anisotropy within the phantom measured, etc.
•	Results section mentioned, “These were sampled from the same biophantom and had similar physical properties”. It does appear that the classification setup is detecting a tissue type where the tissue type is from one sample is present in both the train and test set. This appears to be problematic and an overstatement in the results
•	Result section mentions “Liver biophantoms with incorportated tumours” … “each of its 25 samples from 3 areas of different tumour pathologies correctly classified” where is the results for this as this is highly relevant.
•	In the results section it’s mentioned that “In light of the model’s ability to accurately generalise to unseen data, including the identification of different levels of pathologies within the same biophantom, it is conceivable that with access to a high-quality in/ex vivo tissue dataset containing labelled healthy and pathological samples, similar features could be extracted.” At least in the current results presented and test setup it does not seem possible to make this conclusion as the identification of different levels of pathologies within the same biophantom does not seem to be reported clearly in a single table or figure.

Minor Comments:
o	In the Technology section it is mentioned “it is assumed that this technology will not damage human tissue and is safe” -> would it be possible to calculate the time duration that would reach the unsafe levels? Could compare with existing standards used for ultrasound and heat effect for instance.
o	In the machine learning- approach explain what are the multiple
o	In the implementation section it is mentioned “balanced dataset”-> what is the data breakdown? Also, is this trained within each group or from all data available? What are the classes?

---

### Official Review · Reviewer_KpcU · 2024-08-14
**Towards Intraoperative Tissue Characterisation with Industrial Precision LiDAR**

**Overall Rating:** 4
**Confidence:** 4

**Other Quality Metrics:**

(a) Clarity of writing - fair;
(b) Clinical Significance - good;
(c) Methodological Novelty - fair;
(d) Experiments and Results - good

**Questions For The Authors:**

-

**Strengths:**

-

**Summary Of The Paper:**

In this study, the authors utilized an hand-held Laser Line Probe to detect the tissue viscoelastic information so as to distinguish cancerous tissue from healthy ones. The results on phantom showed good performance.

**Weaknesses:**

1) The structure and writing of this paper can be improved; there is no discussion; the conclusion and future work can be combined;
2) There is no in-vivo validation;
3) Metrics except for accuracy should be provided

---

### Official Review · Reviewer_57NE · 2024-08-14
**Towards Intraoperative Tissue Characterisation with Industrial Precision LiDAR**

**Overall Rating:** 7
**Confidence:** 3

**Other Quality Metrics:**

1) Clarity of writing: Good
2) Clinical Significance: Excellent
3) Methodological Novelty: Excellent
4) Experiments and Results: Good

**Questions For The Authors:**

1) The evaluation would be more informative if it included additional metrics such as AUC-ROC, F1-score, and precision.
2) It is mentioned that data outliers were removed in the scanning step,  but the criteria for these outliers are not clearly explained (which data point is considered as outliers and why). It is important that you mention the criteria to understand this step.

**Strengths:**

1) Introduction of a new dataset
2) Introduction of a new tool
3) Methodology well explained

**Summary Of The Paper:**

This paper investigates the use of an industrial precision handheld Laser Line Probe (LLP) to extract viscoelastic tissue information for classifying healthy and cancerous tissue in real-time. This approach aims to contribute with limited resources and technology.

**Weaknesses:**

Only an accuracy metric is used to evaluate the performance.

---

### Decision · Program_Chairs · 2024-09-23

Accept